# Prevalence and factors associated with mother and newborn skin-to-skin contact in Afghanistan

Essa Tawfiq[1], Muhammad Haroon Stanikzai [2]*, Massoma Jafari[3], Zarghoon Tareen[4], Sayed Ali Shah Alawi[5], Zainab Ezadi[6], Abdul Wahed Wasiq[7], Omid Dadras[8]

1 The Kirby Institute, UNSW Sydney, Sydney, Australia, 2 Department of Public Health, Faculty of Medicine, Kandahar University, Kandahar, Afghanistan, 3 Department of Health Profession Education Research, University of Toronto, Ontario, Canada, 4 Department of Pediatrics, Faculty of Medicine, Kandahar University, Kandahar, Afghanistan, 5 UHI Project/JHPIEGO, Kabul, Afghanistan, 6 Master of Science in Midwifery, Reproductive Health, Kabul, Afghanistan, 7 Department of Internal Medicine, Faculty of Medicine, Kandahar University, Kandahar, Afghanistan, 8 Research Centre for Child Psychiatry, University of Turku, Turku, Finland

* haroonstanikzai1@gmail.com

## Abstract

### Background

Mother-newborn skin-to-skin contact (SSC) involves placing the naked infant on the mother's bare chest within the first hour of birth and is crucial for thermoregulation, bonding, breastfeeding initiation, and promoting neonatal health. This study examined the prevalence, and factors associated with SSC in Afghanistan.

### Methods

Data from the Afghanistan Multiple Indicator Cluster Survey (MICS) 2022–23 were used and analysed from ever-married women, aged 15–49 years, who delivered a live infant in the past 2 years. The outcome was SSC, placing the naked infant on the mother's bare chest and initiating breastfeeding within the first hour of birth. Adjusted odds ratios [AOR: (95%CI)] of factors associated with SSC were obtained by a logistic regression model.

### Results

Of 11,992 women, 32.9% practiced SSC. The likelihood of SSC was greater in women with primary [1.38 (1.14–1.68)] and secondary or higher [1.29 (1.06–1.57)] education, in women who had access to media [1.36 (1.11–1.65)], and those who owned mobile phones [1.27 (1.11–1.45)]. The likelihood of SSC was lower in women who delivered at home [0.26 (0.21–0.33)], those who delivered at private clinics or hospitals [0.50 (0.41–0.61)], and those with cesarean section [0.12 (0.08–0.17)]. Women living in rural areas, and women with deliveries conducted by traditional birth attendants/community healthcare workers and by relatives/others had lower odds of SSC [0.76 (0.63–0.92), 0.37 (0.27–0.53), 0.45 (0.33–0.59), respectively].

**Data availability statement:** The data underlying the results presented in the study are available from the following link: https://mics.unicef.org/surveys?display=card&keys=Afghanistan

**Funding:** The author(s) received no specific funding for this work.

**Competing interests:** The authors have declared that no competing interests exists.

## Conclusion

The low prevalence of SSC in Afghanistan highlights the need for targeted health interventions. Efforts should focus on improving access to public clinics and hospitals, enhancing education, training of healthcare providers, and leveraging media and mobile phone access to promote SSC. Interventions should prioritize rural women and women who have undergone cesarean sections to increase SSC rates and improve neonatal health outcomes.

## Introduction

The first 28 days of a newborn's life are crucial, accounting for 47% of all deaths among children under five worldwide [1]. These deaths are often due to illnesses related to inadequate quality care at birth or the lack of skilled care and treatment immediately after birth and during the first few days [1]. Given this high risk of mortality, immediate interventions such as mother and newborn skin-to-skin contact (SSC) are vital for improving newborn health outcomes.

SSC involves placing the newborn naked (or in a diaper) on the mother's bare chest [2]. The importance of SSC within the first hour of childbirth is to promote initiation of early breastfeeding, and the World Health Organization (WHO) recommends that SSC should be immediate after birth and be continued uninterrupted for at least 60 minutes to support mothers to initiate breastfeeding as soon as possible after birth [3]. This practice promotes early breastfeeding initiation, improves thermoregulation, and enhances neonatal survival while boosting maternal confidence [3–5]. SSC triggers vital reflexes in newborns, especially those born preterm or via cesarean section [3,5], and facilitates the delivery of colostrum, a highly nutritious first milk that contains essential antibodies and immune-boosting substances [3]. For mothers, SSC supports early placenta expulsion [6,7], reduces bleeding [7], lowers stress levels [8], and strengthens mother-infant bonding, which is mediated by increased oxytocin levels [9]. Additionally, SSC reduces the risk of neonatal hypoglycemia and the need for admission to intensive care units [10]. Recent WHO studies highlight that starting kangaroo mother care (KMC) with SSC and exclusive breastfeeding immediately after birth greatly improves survival rates of preterm or low-weight infants, and could prevent up to 150,000 infant deaths annually [11].

In Afghanistan, neonatal mortality remains alarmingly high, with 34 deaths per 1,000 live births, primarily due to prematurity (38%) and birth asphyxia (19%) [12]. Despite the proven benefits of SSC, its practice may be limited in Afghanistan, likely due to cultural and systemic barriers [13]. Cultural norms emphasizing modesty may deter mothers from engaging in SSC, as exposing the chest is deemed inappropriate [14,15]. This cultural pressure can cause SSC to be viewed as inappropriate or unacceptable, limiting its adoption [16]. The lack of privacy in healthcare facilities, coupled with overcrowded conditions, further limits SSC adoption [17]. Similarly, during home births, the presence of male relatives and lack of privacy inhibit mothers from practicing SSC [13,14].

Additionally, there may be limited awareness among healthcare providers and mothers about the importance of SSC for newborns, which hinders its widespread adoption [18]. Challenges within the healthcare system, such as limited resources, lack of training, and high patient-to-provider ratios, further complicate the implementation of SSC [18,19]. The 2016 Afghanistan National Maternal and Newborn Health Quality of Care Assessment reported that while 85% of newborns were dried immediately after birth, only 52% were placed in SSC, and just 62% of skilled birth attendants included thermal protection in essential newborn care [20]. This is particularly concerning as hypothermia is a major contributor to neonatal mortality [21].

Afghanistan is striving to meet the Sustainable Development Goal (SDG) target of reducing neonatal mortality to 12 deaths per 1,000 live births by 2030. To reach this goal, the country must substantially improve essential newborn care, including enhancing the practice of SSC. The latest Multiple Indicator Cluster Survey (MICS) 2022–23 reported that only 15.8% of newborns in Afghanistan receive SSC for thermal care [22]. This low prevalence underscores the urgent need to investigate the factors influencing SSC and re-evaluate its practice within the country.

Given Afghanistan's ongoing socio-political instability, including the increasing humanitarian crises and forced repatriation of Afghan refugees from Pakistan and Iran [23,24], maternal and neonatal health attracts enormous attention because they are at the highest risk of morbidities and mortalities in such situations [25–27]. In this context, SSC within the first hour of delivery is an essential component of maternal and neonatal care [3]. However, research on SSC in Afghanistan remains limited. This study, therefore, seeks to examine the prevalence, and factors associated with SSC in the Afghan context.

## Methods

### Study design and data source

This cross-sectional study used data from the Afghanistan MICS 2022–2023, accessed on June 10, 2024 [22]. The MICS survey covers a wide range of indicators related to the situation of women and children, including child mortality, maternal and newborn health, and water and sanitation. Detailed descriptions of the survey design, sampling methods, and data collection are described elsewhere [22]. During the implementation of MICS, trained surveyors collected data from women of reproductive age (15–49 years old) [22]. In this study, we used and analysed data from 11,992 ever-married women who delivered a live infant in the past 2 years prior to the MICS survey (Fig 1).

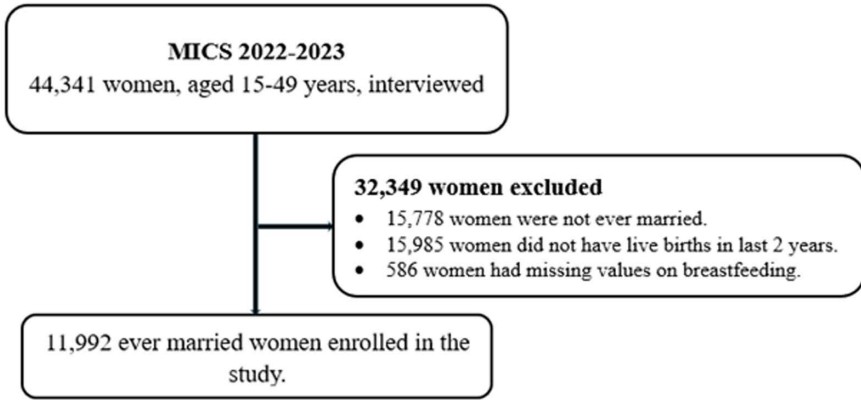

**Fig 1.  Study flowchart.**

## Study variables

The **outcome** was SSC, defined as placing the newborn on the mother's bare chest and initiating breastfeeding within the first hour of birth. SSC was categorized as a binary outcome, coded as "yes" if the mother responded affirmatively to both of the following questions: whether the infant was placed on her bare chest after delivery, and whether breastfeeding was initiated within the first hour of birth. If the response to either question was "no," the outcome was coded as "no."

The **explanatory variables** were selected after a comprehensive literature review [10,28–33]. It includes maternal age (15–19, 20–24, 25–29, 30–39, 40–49 years), maternal education (no formal education, primary education, secondary/higher education), birth order (first child vs. second and more children), infant sex (male vs. female), household wealth status (lowest quintile up to highest quintile), residential area (urban vs. rural), place of delivery (public clinic/hospital, private clinic/hospital, and at home), type of delivery provider (doctor, midwife/nurse, traditional birth attendants (TBAs)/community healthcare worker (CHWs), relatives/other), cesarean section ("yes" if the delivery was by cesarean section, and "no" otherwise), small-size baby ("yes" if the size of the baby was reported to be very small by the interview mother, and "no" otherwise), antenatal care (ANC) visits (no visit, 1–3 visits, 4–7 visits, and ≥ 8 visits), woman used a mobile phone at least once a week in the last 3 months (yes/no), and access to media (yes/no). Access to media was defined as daily TV watching, radio listening, or newspaper reading (yes/no).

## Statistical analysis

Descriptive statistics were used to assess the distribution of sociodemographic characteristics among the study population. The chi-square test was used to examine the relationship between explanatory variables and SSC status. Univariate and multivariable logistic regression models were fitted and run to study the likelihood of SSC across the categories of explanatory variables. In the multivariable model, explanatory variables were selected based on the theoretical relevance after a thorough literature review [10,28–33], and the strength of their relationship with the outcome, was determined by the corresponding $p$-value. The $p$-values obtained from the univariate analysis were used to determine which explanatory variables to include in the multivariate model. Variables with a $p$-value of < 0.25 were added to the multivariate model. In the univariate analysis, women's age and infant's sex had $p$-values > 0.25; therefore, all other explanatory variables were added to the multivariate model except for these two variables. Odds ratio and 95% CI [OR (95%CI)] were obtained for the logistic regression analyses. All analyses accounted for sampling design and weight by defining the survey strata, primary sampling unit, and weight for all analyses in the study. The significant statistical level was set at $p < 0.05$. STATA version 18 was applied for data analyses [34].

## Ethical approval

The study was reviewed by the Research and Ethics Committee, Faculty of Medicine, Kandahar University, Afghanistan. The committee waived the ethical approval (Dated; 20/May/2024) because secondary data from the Multiple Indicator Cluster Survey (MICS) 2022–2023 were used and analyzed in this study.

## Results

As shown in Table 1, 32.9% of women practiced SSC. Of 11,992 women, over 6% were 15–19 years, and over 7% were 40–49 years of age. In terms of age, there was no statistical difference between women who practiced SSC and those who did not. However, statistically significant differences were observed between the groups of women in terms of education, birth order, wealth status, place of delivery, deliveries conducted by type of health provider, cesarean section, ANC visits, access to media, and mobile phone ownership. These differences were in favor of women who practiced SSC compared to women who did not. For example, the proportions of women with secondary or higher education were 18.7% vs. 10.9% for women who practiced SSC and for women who did not (details in Table 1).

**Table 1. Baseline characteristics of ever-married women, by status of skin-to-skin contact.**

| Characteristics | Categories | Whether skin-to-skin contact was practiced | | | p-value |
|---|---|---|---|---|---|
| | | Total<br>n = 11,992 | Yes<br>n = 3,944<br>(32.9%) | No<br>n = 8,048 (67.1%) | |
| Mothers' age | 15–19 years | 6.1% | 6.0% | 6.1% | 0.74 |
| | 20–24 years | 24.8% | 24.8% | 24.9% | |
| | 25–29 years | 29.7% | 30.3% | 29.4% | |
| | 30–39 years | 32.3% | 32.4% | 32.3% | |
| | 40–49 years | 7.1% | 6.6% | 7.4% | |
| Mothers' education | No formal education | 75.9% | 68.3% | 80.1% | <0.001 |
| | Primary | 10.5% | 13.1% | 9.1% | |
| | Secondary/higher | 13.6% | 18.7% | 10.9% | |
| Birth order | 1st child | 16.0% | 17.4% | 15.3% | 0.03 |
| | 2nd or more children | 84.0% | 82.6% | 84.7% | |
| Infant sex | Male | 51.6% | 51.6% | 51.5% | 0.94 |
| | Female | 48.4% | 48.4% | 48.5% | |
| Wealth status | Lowest quintile | 20.7% | 13.4% | 24.7% | <0.001 |
| | Second | 20.9% | 17.7% | 22.7% | |
| | Middle | 20.4% | 20.1% | 20.6% | |
| | Fourth | 19.5% | 23.7% | 17.2% | |
| | Highest quintile | 18.4% | 25.2% | 14.7% | |
| Residential area | Urban | 23.2% | 32.0% | 18.4% | <0.001 |
| | Rural | 76.8% | 68.0% | 81.6% | |
| Place of delivery | Public clinic/hospital | 56.0% | 78.8% | 43.6% | <0.001 |
| | Home | 33.5% | 11.1% | 45.7% | |
| | Private clinic/hospital | 10.5% | 10.2% | 10.6% | |
| Type of delivery provider | Doctor | 9.1% | 12.3% | 7.3% | <0.001 |
| | Midwife/nurse | 53.9% | 73.0% | 43.6% | |
| | TBAs/CHWs | 11.8% | 4.1% | 16.0% | |
| | Relatives/other | 25.2% | 10.6% | 33.2% | |
| Cesarean section | No | 94.4% | 97.9% | 92.5% | <0.001 |
| | Yes | 5.6% | 2.1% | 7.5% | |
| Small-size baby | No | 85.2% | 86.2% | 84.7% | 0.21 |
| | Yes | 14.8% | 13.8% | 15.3% | |
| Antenatal care (ANC) visit | No visit | 22.1% | 15.4% | 25.8% | <0.001 |
| | 1–3 visits | 44.5% | 45.4% | 44.0% | |
| | 4–7 visits | 26.4% | 30.7% | 24.0% | |
| | ≥8 visits | 7.1% | 8.5% | 6.3% | |
| Access to mobile phone | No | 63.1% | 53.8% | 68.2% | <0.001 |
| | Yes | 36.9% | 46.2% | 31.8% | |
| Access to media | No | 80.0% | 72.0% | 84.0% | <0.001 |
| | Yes | 20.0% | 28.0% | 16.0% | |

**Abbreviations:** TBAs, Traditional birth attendants; CHWs, Community healthcare workers.

 presents the likelihood of SSC from bivariate and multivariate analyses. Results from the multivariate analysis show that women's education level was positively associated with SSC: women with primary education, and women with secondary or higher education had higher odds of practicing SSC, compared to women with no formal education [1.38 (1.14–1.68), 1.29 (1.06–1.57), respectively]. Similarly, women with access to media and women with ownership of mobile

**Table 2. Likelihood of skin-to-skin contact practiced by ever-married women.**

| Characteristics | Categories | Crude odds ratio (95% CI) | p-value | Adjusted odds ratio (95% CI) | p-value |
|---|---|---|---|---|---|
| Mothers' age | 15–19 years | Ref | | Ref | |
| | 20–24 years | 1.02 (0.81–1.27) | 0.89 | – | – |
| | 25–29 years | 1.05 (0.84–1.31) | 0.68 | – | – |
| | 30-39 years | 1.02 (0.81–1.28) | 0.87 | – | – |
| | 40–49 years | 0.90 (0.68–1.20) | 0.49 | – | – |
| Mothers' education | No formal education | Ref | | Ref | |
| | Primary | 1.69 (1.41–2.04) | <0.001 | 1.38 (1.14–1.68) | 0.001 |
| | Secondary/higher | 2.02 (1.67–2.43) | <0.001 | 1.29 (1.06–1.57) | 0.01 |
| Birth order | 1st child | Ref | | Ref | |
| | 2nd or more children | 0.86 (0.75–0.99) | 0.03 | 0.98 (0.84–1.14) | 0.79 |
| Infant sex | Male | Ref | | Ref | |
| | Female | 1.00 (0.91–1.09) | 0.94 | – | – |
| Wealth status | Lowest quintile | Ref | | Ref | |
| | Second | 1.44 (1.19–1.74) | <0.001 | 1.09 (0.90–1.33) | 0.36 |
| | Middle | 1.80 (1.50–2.16) | <0.001 | 1.07 (0.89–1.30) | 0.51 |
| | Fourth | 2.54 (2.06–3.13) | <0.001 | 1.15 (0.91–1.45) | 0.24 |
| | Highest quintile | 3.16 (2.54–3.94) | <0.001 | 1.05 (0.79–1.40) | 0.75 |
| Residential area | Urban | Ref | | Ref | |
| | Rural | 0. 48 (0.41–0.56) | <0.001 | 0.76 (0.63–0.92) | 0.006 |
| Place of delivery | Public clinic/health post | Ref | | Ref | |
| | Home | 0. 13 (0.11–0.16) | <0.001 | 0.26 (0.21–0.33) | <0.001 |
| | Private clinic/hospital | 0.53 (0.43–0.65) | <0.001 | 0.50 (0.41–0.61) | <0.001 |
| Type of delivery provider | Doctor | Ref | | Ref | |
| | Midwife/nurse | 1.00 (0.80–1.26) | 0.97 | 0.87 (0.70–1.09) | 0.22 |
| | TBAs/CHWs | 0.15 (0.11–0.22) | <0.001 | 0.37 (0.27–0.53) | <0.001 |
| | Relatives/other | 0.19 (0.15–0.24) | <0.001 | 0.45 (0.33–0.59) | <0.001 |
| Cesarean section | No | Ref | | Ref | |
| | Yes | 0.27 (0.19–0.39) | <0.001 | 0. 12 (0.08–0.17) | <0.001 |
| Small size baby | No | Ref | | Ref | |
| | Yes | 0. 89 (0.74–1.07) | 0.21 | 0.94 (0.77–1.15) | 0.56 |
| Antenatal care (ANC) visits | No visit | Ref | | Ref | |
| | 1–3 visits | 1.73 (1.49–2.02) | <0.001 | 0.97 (0.82-1.14) | 0.68 |
| | 4–7 visits | 2.15 (1.81–2.56) | <0.001 | 0.96 (0.80–1.15) | 0.62 |
| | ≥8 visits | 2.29 (1.47–2.97) | <0.001 | 1.06 (0.80–1.40) | 0.71 |
| Access to mobile phone | No | Ref | | Ref | |
| | Yes | 1.84 (1.63–2.07) | <0.001 | 1.27 (1.11–1.45) | <0.001 |
| Access to media | No | Ref | | Ref | |
| | Yes | 1.96 (1.65–2.33) | <0.001 | 1.36 (1.11–1.65) | 0.003 |

**Abbreviations:** TBA, Traditional birth attendants; CHW, Community healthcare workers.

phones had higher odds of practicing SSC [1.36 (1.11–1.65), and 1.27 (1.11–1.45)], compared to women with no access to media and women with no ownership of mobile phones, respectively. The likelihood of SSC was lower in women who delivered at home [0.26 (0.21–0.33)], and in those delivered at private clinics or hospitals [0.50 (0.41–0.61)], compared to those delivered at public clinics or hospitals. Women who had cesarean section deliveries were less likely to practice SSC compared to those with vaginal deliveries [0.12 (0.08–0.17)]. Likewise, women living in rural areas, compared to those living in urban areas, women with deliveries conducted by TBAs/CHWs, and by relatives/other, compared to women whose deliveries were conducted by doctors, had lower odds of practicing SSC [0.76 (0.63–0.92, 0.37 (0.27–0.53), and 0.45 (0.33–0.59), respectively].

## Discussion

This study found that only a third of Afghan women practiced SSC. The likelihood of SSC was higher in women with primary or higher education, and those with access to media and mobile phones. On the contrary, the likelihood of SSC was lower in women who delivered at home, those who delivered at private clinics or hospitals, women with deliveries conducted by TBAs/CHWs and by relatives/others, rural women, and those who had cesarean sections.

The SSC prevalence of 32.9% observed in this study is double that reported by the Afghanistan MICS 2022–23 [22]. This discrepancy may be attributed to differences in our calculation of SSC prevalence, as we excluded women who did not have a live birth in the past 2 years. The prevalence of SSC practice is lower than the 52% reported previously in Afghanistan [20]. Considering the importance of SSC within the first hour after childbirth and initiation of early breastfeeding, our definition of SSC covered the placement of the naked newborn on the mother's bare chest and initiation of breastfeeding within 60 minutes of the delivery. Nevertheless, our prevalence aligns with findings from other low- and middle-income countries (LMICs), where SSC rates range from 28.0% to 45.7% [30,35,36]. The low SSC prevalence remains a pressing concern, emphasizing the need for targeted efforts to improve SSC practices in Afghanistan. The factors identified in our study should guide future interventions and policy developments.

This study showed that SSC prevalence was significantly higher among women with primary or higher education levels compared to those with no formal education. This is consistent with previous research in LMICs, where maternal education has been linked to a better understanding of neonatal care and improved SSC rates [30,35,37]. Our findings underscore the critical importance of women's education, particularly in light of the recent restrictions on female education in Afghanistan [38]. The findings of this study reinforce the need to support continued advocacy for women's education in Afghanistan. Additionally, public health campaigns aimed at raising awareness of SSC among less-educated women are urgently needed.

We found that women living in rural areas were less likely to practice SSC than those living in urban areas. This finding is consistent with the studies conducted in Ethiopia [36,39], and Nigeria [32]. Women living in rural areas tend to have poor access to maternal and child healthcare services, which could explain the observed association [32]. Moreover, earlier studies in Afghanistan [40–43], revealed that rural women are at a disadvantage in accessing maternal and child healthcare services. Therefore, it is essential to design interventions that specifically target this population. Policymakers must prioritize improving healthcare accessibility for rural mothers and newborns to address these inequalities.

Consistent with findings from other LMICs, our study found a positive association between deliveries at health facilities and SSC [28,30]. Access to maternal and neonatal healthcare in public health facilities can improve breastfeeding practices, including early initiation of breastfeeding [44], SSC [30], and exclusive breastfeeding [45]. Compared to private facilities, public healthcare facilities in Afghanistan are more closely aligned with national health policies and more frequently receive support and training from international partners to enhance maternal and newborn health services, including SSC practices [15,18,25,46]. In this study, we also observed that women whose deliveries were conducted by TBAs/CHWs and relatives/others were less likely to practice SSC. Therefore, increasing access to institutional deliveries

may be a necessary policy consideration for improving SSC practices in Afghanistan [47]. Moreover, the findings highlight the need for targeted qualitative improvement initiatives, continuous training of healthcare workers, and public awareness efforts to ensure consistent SSC practices in Afghanistan. In addition to education for healthcare workers, public awareness campaigns may help to educate TBA/CHWs and relatives/others about the importance of SSC which may contribute to consistent SSC practices in Afghanistan.

Another important finding is the negative association of cesarean sections with SSC practices in our study, which corroborates with findings from previous studies [28,30]. A plausible reason could be the delay in initiating breastfeeding due to the recovery of the mother post-cesarean section operation. Additionally, the condition of the newborn after a cesarean section can also impact SSC practices [28]. Newborns delivered via cesarean section may experience distress or other complications, such as respiratory issues, which can necessitate immediate medical attention and further delay SSC [28,35]. This finding suggests that targeted interventions to improve SSC for newborns delivered by cesarean section are needed at health facilities, given WHO emphasizes supporting the dyad to engage in SSC practice, regardless of the mode of delivery [3]. The evidence from a literature review suggests that with appropriate collaboration of healthcare workers in maternity wards, SSC during cesarean section surgery can be effectively implemented [48]. To improve SSC after cesarean sections, healthcare facilities can assign a dedicated staff member to monitor SSC, ensuring newborn safety and supporting maternal bonding [49]. Staff education on SSC benefits—such as promoting breastfeeding, calming the newborn, and reducing stress—can strengthen team commitment [50]. Adjusting operating room protocols, like positioning the newborn on the mother's chest with warm blankets, can also facilitate SSC [49,50]. Institutional policies should make SSC a standard practice to minimize mother-infant separation, aligning with WHO and UNICEF recommendations [51].

Data from LMICs consistently highlight the strong association between maternal mobile ownership and improved newborn care, such as better child feeding and timely immunization [52,53]. Similarly, we found that women with access to mobile phones were 1.2 times as likely to practice SSC compared to those with no access to mobile phones. Access to mobile phones likely enhances exposure to health education on breastfeeding and maternal and child health, as seen in other LMICs [53,54]. Recent evidence from Afghanistan identifies mobile health (mHealth) technologies as a potential opportunity to promote maternal and newborn health [55,56]. Thus, mHealth interventions may potentially enhance SSC practices, and may also contribute to improving other maternal healthcare utilization and newborn care indicators in the country.

Finally, consistent with studies from sub-Saharan Africa [28] and Nigeria [57], mothers' access to media was strongly associated with SSC practice; e.g., women with access to media were about 1.3 times as likely to practice SSC than those with no access to media. Media plays an essential role in disseminating information on newborn care, particularly to less-educated women [58,59]. Educational campaigns through various media platforms have proven effective in improving maternal healthcare utilization and SSC practices in LMICs [58,60]. Therefore, expanding media-based education on maternal and child health in Afghanistan should be prioritized.

## Limitations

This study has some limitations. First, recall bias may have affected the accuracy of the data, as women were asked to report events that took place several months ago. Moreover, there were slightly over 1% (131 observations) with responses of "don't know" for the SSC outcome, and this percent of responses were coded as "no SSC" for the outcome. This could have led to either under or over-reported SSC practice. Second, the MICS did not collect data on mother's knowledge of breastfeeding; thus, we couldn't examine the association between this important predictor and SSC practice. Furthermore, we were unable to account for other important predictors of SSC practice such as healthcare provider training and attitudes, prenatal breastfeeding intention, counseling on breastfeeding during ANC visits, cultural and social norms, maternal mental health, support from family or birth attendants, and previous birth experience. Hence, further research is warranted to examine them.

## Conclusion

Health interventions should be designed and implemented to address the low prevalence of SSC in Afghanistan. These interventions should focus on improving access to public healthcare facilities, enhancing educational opportunities for women, training healthcare providers, and promoting the use of media and mobile phones for health education. Special attention should be given to rural women and women who undergo cesarean sections. Prioritizing these groups is essential to increasing SSC rates and improving maternal and neonatal health outcomes in Afghanistan.

## Supporting information

**S1 File. Human_Subjects_Research_Checklist.**
(PDF)

## Acknowledgments

We thank the UNICEF for allowing us to access and analyze this data.

## Author contributions

**Conceptualization:** Essa Tawfiq, Muhammad Stanikzai, Zarghoon Tareen, Abdul Wahed Wasiq, Omid Dadras.

**Data curation:** Essa Tawfiq, Muhammad Stanikzai, Zainab Ezadi, Omid Dadras.

**Formal analysis:** Essa Tawfiq, Muhammad Stanikzai, Omid Dadras.

**Investigation:** Essa Tawfiq, Muhammad Stanikzai, Omid Dadras.

**Methodology:** Essa Tawfiq, Muhammad Stanikzai, Massoma Jafari, Sayed Ali Shah Alawi, Abdul Wahed Wasiq, Omid Dadras.

**Project administration:** Muhammad Stanikzai.

**Resources:** Muhammad Stanikzai.

**Software:** Muhammad Stanikzai.

**Supervision:** Muhammad Stanikzai.

**Validation:** Muhammad Stanikzai.

**Visualization:** Muhammad Stanikzai.

**Writing – original draft:** Essa Tawfiq, Muhammad Stanikzai, Massoma Jafari, Zarghoon Tareen, Sayed Ali Shah Alawi, Zainab Ezadi, Omid Dadras.

**Writing – review & editing:** Essa Tawfiq, Muhammad Stanikzai, Massoma Jafari, Zarghoon Tareen, Sayed Ali Shah Alawi, Zainab Ezadi, Abdul Wahed Wasiq, Omid Dadras.

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
