## [Decision Letter · Decision Letter 0]

1 Nov 2024

PONE-D-24-41027Prevalence and factors associated with mother and newborn skin-to-skin contact in AfghanistanPLOS ONE

Dear Dr. Stanikzai,

Thank you for submitting your manuscript to PLOS ONE. After careful consideration, we feel that it has merit but does not fully meet PLOS ONE’s publication criteria as it currently stands. Therefore, we invite you to submit a revised version of the manuscript that addresses the points raised during the review process.

We look forward to receiving your revised manuscript.

Kind regards,

Md. Moyazzem Hossain, PhD

Academic Editor

PLOS ONE

Journal Requirements:

Reviewers' comments:

Reviewer's Responses to Questions

**Comments to the Author**

1. Is the manuscript technically sound, and do the data support the conclusions?

Reviewer #1: Yes

Reviewer #2: Yes

Reviewer #3: Partly

2. Has the statistical analysis been performed appropriately and rigorously? 

Reviewer #1: Yes

Reviewer #2: Yes

Reviewer #3: I Don't Know

3. Have the authors made all data underlying the findings in their manuscript fully available?

Reviewer #1: Yes

Reviewer #2: Yes

Reviewer #3: Yes

4. Is the manuscript presented in an intelligible fashion and written in standard English?

Reviewer #1: Yes

Reviewer #2: Yes

Reviewer #3: Yes

5. Review Comments to the Author

Reviewer #1: Skin-to-skin care (SSC), also known as kangaroo care, is a well-established evidence-based practice that has been shown to improve neonatal outcomes. This study investigates the prevalence and barriers to SSC implementation. The findings are significant and support the continued promotion of SSC, the highlighted interventions are reasonable and feasible.

Additional references are needed to strengthen the evidence base. (references are needed in lines 86, 91, 200, and 249).

Lines 211-212 is not needed here, already mentioned more than once in the manuscript. Please delete.

The discussion section is redundant, and could be condensed to focus on the key findings and implications.

Were there any differences in the likelihood of SSC based on the gender of the delivering provider (physician, nurse, midwife, etc.)?

Did mothers who received breastfeeding counseling during pregnancy have a different likelihood of engaging in SSC?

Reviewer #2: Thank you for the opportunity to peer review this important research regarding skin-to-skin contact and early breastfeeding initiation. The manuscript is written in a concise manner with strong supporting rationale.

I would like to offer some suggestions for consideration:

Line 115- Can you add some details about the MIICS such as number of questions and type of questions (i.e. multiple choice, how it was administered)? In which languages was the survey available? Etc.

Line 117- suggest use the word “live infant” instead of “child”

Line 126 - Suggest to analyze adolescents separately (19 years of age and less). This will make the groups more balanced and helpful to identify targeted education interventions. I see in Table TM 4.2 they have broken down age categories, and list as "less than 20" years of age (p. 75) https://mics.unicef.org/sites/mics/files/Afghanistan%202022-23%20MICS_English.pdf

Line 128- “infant sex” instead of child; “male vs female” instead of boy vs girl

Line 130 - suggest to spell out cesarean instead of c-section. Also suggest to reword the sentence “(“yes” if the delivery was by cesarean…”

Line 131-132- the “small-size baby” category seems highly subjective. Was there any data on birth weight? Can you report SGA or LBW instead?

Line 133- What's the reference for these categories of antenatal care visits? Four visits would not be adequate prenatal care as defined by other authors. There is certainly lots of debate about what is considered to be adequate. See recent integrative review https://onlinelibrary.wiley.com/doi/10.1111/jmwh.13459

In the Unicef data summary, WHO is cited, with a minimum of 8 antenatal care visits recommended. (p. 72) https://mics.unicef.org/sites/mics/files/Afghanistan%202022-23%20MICS_English.pdf

Suggest reanalyzing the data including the following categories of number of antenatal visits: 4-7, 8 or more.

Line 138- correction – it is chi-square

Line 146- add citation for STATA. What variables did you use when running the adjusted analysis? You need to add this.

Line 156- add the word respectively to the end of the sentence ending in SSC.

Line 168 - will need to recalculate if using different age group categories as previously suggested.

Line 176- suggest “vaginal” instead of normal deliveries

Table 1. Suggest to include standardized differences with a standardized difference of 0.10 or greater indicating an imbalance in groups (instead of p-values). Include a legend at the bottom of the table or spell out ANC in table.

Line 196-197. Suggest to restate the conclusion for women from higher socioeconomic backgrounds. The second quintile is not much higher than the first quintile. The other quintiles were non-significantly associated in the adjusted analysis

Line 202 – Re inclusion criteria of live birth within the last 2 years. Does this finding suggest that there may be an increasing trend in the use of SSC? Have there been any policy changes or public health campaigns that might have contributed to this? Certainly you reported that public clinics/hospitals were supporting this practice more so than private clinics/hospitals. Suggest to elaborate.

Line 219- Suggest the following sentence “The findings of this study reinforce the need to support continued advocacy for women's education in Afghanistan”

Line 222- Suggest “Consistent” instead of “in line with…”

Line 224-225- Would this also be related to receiving health education about the importance of SSC and early breastfeeding initiation?

Line 245- Suggest to include citation to the International Childbirth Initiative: 12 Steps to Safe and Respectful Mother-Baby Family Maternity Care. https://icichildbirth.org/initiative/

Line 253- Suggest “WHO emphasizes supporting the dyad to engage in SSC…”

Line 256- Suggest to describe your recommendation in more detail.

Often transient tachypnea of the newborn can be resolved by placing the newborn on the mother's chest, with close observation of the newborn. Perhaps this is an education component for staff. Someone on the team needs to be assigned to the baby for safety reasons (to ensure safe postnatal adaptation).

Can also cite the Global Breastfeeding Collective Advocacy Brief. https://www.globalbreastfeedingcollective.org/media/376/file/Breastfeeding%20in%20emergency%20situations.pdf

Line 269- 1.3 times as likely

Line 28- 1.2 times as likely

Line 288- Also, prenatal breastfeeding intention is important as well

I wish you great success in your continued efforts to improve care for mother-baby dyads.

Reviewer #3: Thank you for the team for their work in producing this manuscript. While I think it’s an important question to understand determinants of SSC and that MICS data is high quality data to utilize for this analysis, I do have some questions or concerns regarding the analytical methodology and rationale behind some considerations.

Primarily, since this utilizes complex survey data, were appropriate weights applied when running the analyses to consider the survey design? With multi-level survey data, descriptive statistical findings will be incorrect without weighting. It was not clear in the methods whether this was done.

In the methods, it should also be clarified which variables were included in the multivariate analysis (was it all of the variables?) This can help a reader contextualize the findings in understanding what you controlled for. Also, the alpha level should be clarified.

For the SSC outcome, were there any ‘don’t know’ responses? How were those coded? This is relevant for an outcome susceptible to recall bias.

What was the rationale behind including only ever-married women? What percentage of mothers are ever-married, and if it is a non-negligible proportion of mothers, might they be qualitatively different from never-married women?

I agree that you included several key variables in your analysis, but I would have also included skilled birth attendance, which can greatly affect the quality of care of home births. You note this in the introduction, and also note that a substantial proportion of births occurred at home. In addition, you should include more detail or references on the literature review that led to the selection of variables.

In the first paragraph of the results section, you should focus on the findings relevant to the outcome of SSC. You write about the characteristics of the whole population, but the relevant text here would be comparisons for how these characteristics differ between those with and without SSC.

I understand the importance of the data in Figure 2, but I am not sure that it is necessary, as it repeats the findings in Table 1.

For Table 2, since you don’t discuss the findings for the univariate models, I would not include those here (perhaps in an appendix). Since the multilevel analysis implies that controlling for other variables are important, it implies that crude estimates are insufficient.

In line 171-172, the findings should not only compare two wealth quintiles (which is a somewhat arbitrary division of the population), but rather speak more to the relationship between wealth and the outcome, perhaps including findings from the other wealth quintiles.

On line 85, I think ‘emphasize’ should be ‘emphasizing’. The discussion and introduction were both well written.

I hope these comments provide useful guidance.

6. PLOS authors have the option to publish the peer review history of their article (what does this mean? ). If published, this will include your full peer review and any attached files.

**Do you want your identity to be public for this peer review?** For information about this choice, including consent withdrawal, please see our Privacy Policy .

Reviewer #1: **Yes: ** Mahmoud AM Ali, MD.

Reviewer #2: No

Reviewer #3: No

---

## [Author Response · Author response to Decision Letter 1]

15 Nov 2024

We have uploaded a response letter.

---

## [Decision Letter · Decision Letter 1]

26 Mar 2025

PONE-D-24-41027R1Prevalence and factors associated with mother and newborn skin-to-skin contact in AfghanistanPLOS ONE

Dear Dr. Stanikzai,

Thank you for submitting your manuscript to PLOS ONE. After careful consideration, we feel that it has merit but does not fully meet PLOS ONE’s publication criteria as it currently stands. Therefore, we invite you to submit a revised version of the manuscript that addresses the points raised during the review process.

We look forward to receiving your revised manuscript.

Kind regards,

Md. Moyazzem Hossain, PhD

Academic Editor

PLOS ONE

Journal Requirements:

Reviewers' comments:

Reviewer's Responses to Questions

**Comments to the Author**

1. If the authors have adequately addressed your comments raised in a previous round of review and you feel that this manuscript is now acceptable for publication, you may indicate that here to bypass the “Comments to the Author” section, enter your conflict of interest statement in the “Confidential to Editor” section, and submit your "Accept" recommendation.

Reviewer #2: All comments have been addressed

2. Is the manuscript technically sound, and do the data support the conclusions?

Reviewer #2: Partly

3. Has the statistical analysis been performed appropriately and rigorously? 

Reviewer #2: No

4. Have the authors made all data underlying the findings in their manuscript fully available?

Reviewer #2: Yes

5. Is the manuscript presented in an intelligible fashion and written in standard English?

Reviewer #2: Yes

6. Review Comments to the Author

Reviewer #2: Thank you for taking into consideration the suggested changes from my review as well as the other reviewers.

I noted that “access to radio” had a p-value of >0.25. Was this an exception to what you had stated in line 152 about the variables that included in your adjusted analysis (i.e. on lines 140-141, “access to media” was a combination of access to TV, radio and newspaper).

I would like to offer some suggestions for consideration:

An important implication of your study findings is that TBA/ CHWs and relatives/others should receive education regarding the importance of SSC and how to do this safely. Not only does SSC help with bonding, but SSC can help with thermoregulation, prevent hypoglycemia, and stabilize respiratory effort among many other benefits. These are very important particularly in rural and remote areas, as you mentioned. I would suggest expanding lines 265-266. In addition to education for healthcare workers, public awareness campaigns may help to educate TBA/CHWs and relatives/others about the importance of SSC which may contribute to consistent SSC practices in Afghanistan.

Re: lines 232-233. Could your findings of a low rate of SSC be related to how you defined SSC? On lines 124-128 you indicated that to be coded a “yes” for SSC, mothers must have said yes to 1) baby was placed on bare chest after delivery and 2) breastfeeding initiated within the first hour of birth. Might this explain the difference in prevalence? Do other researchers using the MICS survey data combine those two actions? To me, they are separate. All babies benefit from SSC. For example, a mother may wish to bottle feed her baby, yet the baby would still benefit from SSC.

I wish you great success in sharing these findings and improving care for women and babies in Afghanistan.

7. PLOS authors have the option to publish the peer review history of their article (what does this mean? ). If published, this will include your full peer review and any attached files.

**Do you want your identity to be public for this peer review?** For information about this choice, including consent withdrawal, please see our Privacy Policy .

Reviewer #2: No

---

## [Author Response · Author response to Decision Letter 2]

12 Apr 2025

We have uploaded a response letter.

---

## [Decision Letter · Decision Letter 2]

2 May 2025

Prevalence and factors associated with mother and newborn skin-to-skin contact in Afghanistan

PONE-D-24-41027R2

Dear Dr. Stanikzai,

We’re pleased to inform you that your manuscript has been judged scientifically suitable for publication and will be formally accepted for publication once it meets all outstanding technical requirements.

Kind regards,

Md. Moyazzem Hossain, PhD

Academic Editor

PLOS ONE

Additional Editor Comments (optional):

Reviewers' comments:

Reviewer's Responses to Questions

**Comments to the Author**

1. If the authors have adequately addressed your comments raised in a previous round of review and you feel that this manuscript is now acceptable for publication, you may indicate that here to bypass the “Comments to the Author” section, enter your conflict of interest statement in the “Confidential to Editor” section, and submit your "Accept" recommendation.

Reviewer #2: All comments have been addressed

2. Is the manuscript technically sound, and do the data support the conclusions?

Reviewer #2: Yes

3. Has the statistical analysis been performed appropriately and rigorously? 

Reviewer #2: Yes

4. Have the authors made all data underlying the findings in their manuscript fully available?

Reviewer #2: Yes

5. Is the manuscript presented in an intelligible fashion and written in standard English?

Reviewer #2: Yes

6. Review Comments to the Author

Reviewer #2: (No Response)

7. PLOS authors have the option to publish the peer review history of their article (what does this mean? ). If published, this will include your full peer review and any attached files.

**Do you want your identity to be public for this peer review?** For information about this choice, including consent withdrawal, please see our Privacy Policy .

Reviewer #2: No

---

## [Editor Report · Acceptance letter]

PONE-D-24-41027R2

PLOS ONE

Dear Dr. Stanikzai,

I'm pleased to inform you that your manuscript has been deemed suitable for publication in PLOS ONE. Congratulations! Your manuscript is now being handed over to our production team.

Kind regards,

on behalf of

Professor Md. Moyazzem Hossain

Academic Editor

PLOS ONE